# Exploring regional disparities in lung cancer mortality in a Brazilian state: A cross-sectional ecological study

Vlaudimir Dias Marques[1]*, Miyoko Massago[1], Mariana Teixeira da Silva[1], Igor Roskowski[1], Daniel Augusto Nunes de Lima[1], Lander dos Santos[1], Estela Louro[1], Simone Tomás Gonçalves[1], Raissa Bocchi Pedroso[1], Armstrong Mbi Obale[2], Sandra Marisa Pelloso[1], João Ricardo Nickenig Vissoci[1,2], Catherine Ann Staton[1,2], Oscar Kenji Nihei[3], Maria Dalva de Barros Carvalho[1], Amanda de Carvalho Dutra[1], Luciano de Andrade[1,2]

1 Postgraduate Program in Health Sciences, State University of Maringa, Maringa, Parana, Brazil, 2 Duke Global Health Institute, Duke University, Durhan, North Carolina, United States of America, 3 Education, Languages and Health Center, Western Parana State University, Foz do Iguaçu, Parana, Brazil

* vdmarques@uem.br

**Data Availability Statement:** https://figshare.com/s/f6a8509b272a8c247a30.

## Abstract

### Background

Lung cancer (LC) is one of the main causes of mortality in Brazil; geographic, cultural, socio-economic and health access factors can affect the development of the disease. We explored the geospatial distribution of LC mortality, and associated factors, between 2015 and 2019, in Parana state, Brazil.

### Methods and findings

We obtained mortality (from the Brazilian Health Informatics Department) and population rates (from the Brazilian Institute of Geography and Statistics [IBGE]) in people over 40 years old, accessibility of oncology centers by municipality, disease diagnosis rate (from Brazilian Ministry of Health), the tobacco production rate (IBGE) and Parana Municipal Performance Index (IPDM) (from Parana Institute for Economic and Social Development). Global Moran's Index and Local Indicators of Spatial Association were performed to evaluate the spatial distribution of LC mortality in Parana state. Ordinary Least Squares Regression and Geographically Weighted Regression were used to verify spatial association between LC mortality and socioeconomic indicators and health service coverage. A strong spatial autocorrelation of LC mortality was observed, with the detection of a large cluster of high LC mortality in the South of Parana state. Spatial regression analysis showed that all independent variables analyzed were directly related to LC mortality by municipality in Paraná.

### Conclusions

There is a disparity in the LC mortality in Parana state, and inequality of socioeconomic and accessibility to health care services could be associated with it. Our findings may help health

**Funding:** The authors received no specific funding for this work.

**Competing interests:** The authors have declared that no competing interests exist.

managers to intensify actions in regions with vulnerability in the detection and treatment of LC.

## Introduction

Lung cancer (LC) is the second most frequent type of cancer worldwide with more than 2.2 million new cases (11.4% of total) in 2020. It is also the second most lethal type of cancer with about 1.8 million deaths (19.4%) in 2020; this is one death for every five patients with LC [1, 2]. In South America, the incidence of LC is 17.8 per 100,000 inhabitants for men and 10.3 per 100,000 inhabitants for women[1]. In Brazil, LC is also the second most common cancer in men and fourth in women [3]. Over 100,000 Brazilians died from lung cancer between 2000 and 2015 [4].

Deaths in Brazil due to lung cancer, between 2000 and 2015, were higher in states with high Human Development Index[4]. Other factors, such as low access to information and health services, geographic location, environmental and occupational exposure, diet, and expensive technologies still continue to pose a challenge to both, the prevention of mortality by LC and to the public or private health systems [5–12]. As a matter of fact, early diagnosis is critical to successful treatment [13].

Brazil is the second largest producer of *Nicotiana tabacum* leaf worldwide. The southern region of the country is responsible for about 96% of the national production of tobacco [14]. Moreover, in southern Brazil and among elderly, there has been higher incidences of neoplasms linked to tobacco consumption, and they are typically found in the mouth, esophagus, and lungs [15, 16]. Smokers do not usually access health services as frequently as nonsmokers [17]. The stigma associated with smoking, including discrimination and prohibition in Brazil of smoking in enclosed spaces, contributes to the poor use of healthcare services [17, 18].

A major move towards establishing new strategies aiming to prevent premature deaths is to understand the spatial distribution of mortality by LC and associated factors, with effective control of risk factors and implementation of appropriate prevention strategies, which has been associated with decrease of neoplasm incidence by more than 30 percent [2]. The characteristics of LC and associated mortality can vary from place to place. However, there is a lack of knowledge on the spatial distribution of LC mortality in municipalities of southern Brazil.

Based on this, the objective of this study was to analyze the regional disparities and factors associated with LC mortality in people over 40 years old residing in the State of Parana, Brazil, between 2015 and 2019; we sought to investigate associations between LC mortality rates and socioeconomic variables, exam coverage rates, and accessibility to health services, by municipality.

## Methods

### Study design and setting

Following the recommendations of the Strengthening the Reporting of Observational Studies in Epidemiology Guideline [19], we developed a cross-sectional ecological study using secondary data of LC mortality that occurred between 2015 and 2019, in Parana state, Brazil. Parana is the fifth most populous state in Brazil. It has a population of 11,516,840 inhabitants, and occupies a surface area of 199,800 km$^2$. Parana is located in the southern region of the country and consists of 399 municipalities divided into 22 regional health (Fig 1). Its latitude and longitude coordinates range from 22˚30'58"to 26˚43'00", and 48˚05'37" to 54˚37'08", respectively [20].

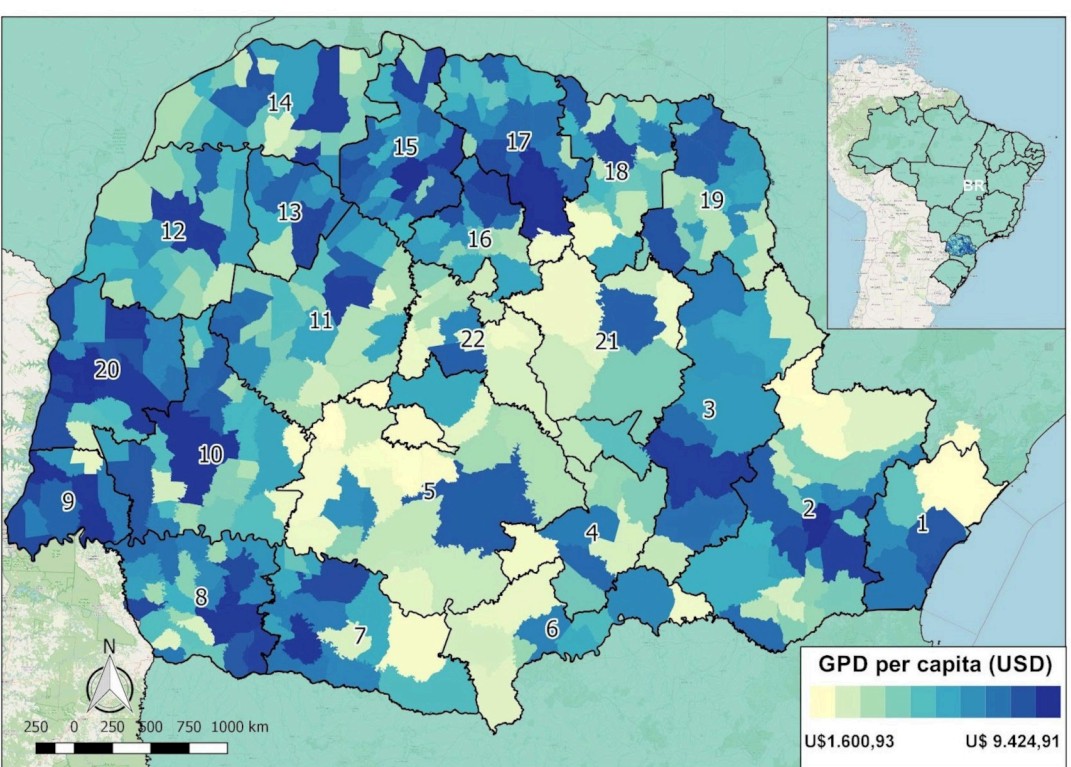

**Fig 1. The cartographic base with location of all municipalities of the state of Parana and its variations in income in 2019 and the state 22 regional health** [24, 25].

In terms of gross domestic product, Parana ranks among the top five in Brazil, despite heterogeneity in the distribution of income and resources, as well as significant economic dynamic differences among the regions [21, 22]. It also has the fifth lowest poverty rate and a high literacy rate, corresponding to a good social and economic condition [22, 23]. In 2020, the estimated prevalence of cancer of the respiratory system in Parana state was 13.81 and 20.78 per 100,000 inhabitants for females and males, respectively [24]. This state is the third largest harvester of *Nicotiana tabacum* in Brazil [15].

## Study variables and sources

**Outcome variable.** Our outcome variable was LC mortality rate. Data about the number of deaths due to LC were obtained from the Mortality Information System of the Brazilian Health Ministry (https://datasus.saude.gov.br/mortalidade-desde-1996-pela-cid-10). The data are coded by the International Statistical Classification of Diseases and Related Health Problems (ICD-10) as C34.9, in people above 40 years old, registered between 2015 and 2019.

The population of each municipality between 2015 and 2019 was obtained from the Brazilian Institute of Geography and Statistics (https://www.ipardes.pr.gov.br/). The LC mortality rate for each municipality was calculated by dividing the number of deaths due to LC by the age-adjusted population for each municipality in the state of Parana. LC data is secondary and freely available with a delay of two years and there is no missing data, but they can have some limitations which are minimized using a big dataset.

**Variables.** To draw a death profile related to the lung cancer mortality rate, we evaluated 4 socioeconomic, demographic and access to health factors according to the patient's city of residence, as described below. The selection of these variables for the final model was based on the low multicollinearity criterion (conditional number test) [26].

Municipal socioeconomic indicators: Parana Municipal Performance Index (IPDM) is an index that measures the performance of the 399 municipalities in the State of Parana, considering three dimensions: income and agricultural production, health, and education. The IPDM data were obtained from the Parana Institute for Economic and Social Development (https://www.ipardes.pr.gov.br/).

Tobacco production rate: TPR (total in tons per planted area [hectares]) was obtained from the Systematic Survey of Agricultural Production of the Brazilian Institute of Geography and Statistics (https://sidra.ibge.gov.br/tabela/1618).

Chest computed tomography rate: The diagnostic exam rates were obtained from the Brazilian Hospital Information System (https://datasus.saude.gov.br/acesso-a-informacao/producao-hospitalar-sih-sus/). The calculation of rates was performed using the number of exams per municipality / population over 15 years old x 1000. All data were acquired using RStudio software (version 4.1.0) [27].

Accessibility to oncology centers index: To calculate the accessibility index we applied the technical procedure called Two Step Floating Catchment Area (2SFCA) through the ArcGIS® software (ESRI) which has two stages, where catchment coverage areas are created for oncology referral centers. In the first step, a catchment area is created that the oncology referral center (provider) can reach in a given distance (for this study 60 Km), then the proportion between provider and population is calculated. In the second step, the coverage of the calculated proportion, in the distance of 60 kilometers, from the centroids of the municipalities, so the index is then generated by summing all provider indices for the population [28, 29]. The data Specialized Medical Services-oncology were obtained from the National Registry of Health Establishments (https://datasus.saude.gov.br/acesso-a-informacao/producao-hospitalar-sih-sus/) [30].

## Spatial analysis

**Spatial autocorrelation.** To reduce the instability, the LC mortality rates were smoothed. We used a Spatial Empirical Bayes estimator of GeoDa software to construct a Queen type binary matrix and obtain the mortality rate per 100,000 inhabitants in each city of Parana [27].

This matrix allowed us to determine neighboring municipalities that have a similar geographic border by measuring the non-random association between the value of a variable in a given geographic unit and the value of variables in neighboring units [31]. Based on the Queen Matrix results and using Global Spatial Autocorrelation Index (Moran's I) and Local Spatial Autocorrelation Index (LISA) statistics [32, 33] a spatial analysis was applied to represent the occurrence of LC mortality spatiality.

Moran's I analysis evaluates if the patterns are globally classified as cluster, scattered or randomly distributed, by calculating the global mean value and the variance of the LC mortality probability, followed by the creation of cross-product (multiplication of all neighboring features standard deviation) [29]. This index varies between -1 (negative autocorrelation) and +1 (positive correlation). If the neighbor's feature shows a higher or smaller result than the mean value, it has a positive cross-product. This means it forms the spatial cluster with the same patterns (high-high or low-low) and Moran's I is positive. Otherwise, if the neighbor's feature has the same value as the mean value, it means there is no cluster [33].

Given that the absence of global autocorrelation does not always indicate the absence of local correlation, the Local Spatial Autocorrelation Index (LISA) was also performed to find possible spatial association patterns by finding local clusters, with a 95% confidence level and significance of 5% (p<0.05) [31]. These clusters were classified into high-high (municipalities with high LC mortality rate surrounded by the others with the same pattern) also named hot spots, and low-low (municipalities with low LC mortality boarded by the others with low same pattern) or cold spots [34].

**Spatial regression.**   Multivariate spatial analysis was performed using the estimation methods. We determined the association between tobacco production rate, Parana Municipal Performance Index, accessibility to oncology centers by municipality, disease diagnosis rate, and LC mortality, the dependent variable. This was done using GeoDa software [35, 36] to perform Ordinary Least Square (OLS) regression and GWR4 software [37, 38] to perform Geographically Weighted Regression (GWR). The OLS uses linear regression and seeks the best model fit by globally comparing the residual differences between the predicted and the actual value for each independent variable and municipality, to find the best match between them. Variables with values lower than -1.96 or higher than 1.96 are considered statistically different [35] and they were used for GWR technique.

Based on low multicollinearity criteria, which did not influence the regression results, GWR evaluates the correlation between dependent and independent variables for each locality. So, this methodology considers spatial dependence, computing separated regression, and identifying local geographical significant clusters within the studied area [37]. For this, a regression equation for each significant index value in OLS is fitted to verify the presence of non-spatial parking for each group of datasets. These equations are constructed separately and they incorporate the dependent feature variables which fall in the neighborhood of target ones [37, 38]. To evaluate the performance of the models, the parameters that were highly related to LC mortality were selected, and the adjusted $R^2$, Akaike Information Criterion (AIC) and Moran's I residues were analyzed based on the classic protocol for multivariate regression model.

**Ethical aspects.**   The data used in this work were publicly available secondary data from government databases, and without any patient identifiers. For this reason, according to the resolution 510 published in 2016 by the Brazilian National Health Council, this study was exempt from approval of the Health Research Committee.

## Results

Between 2015 and 2019, 9016 adults over the age of 40 died from LC in the Brazilian state of Parana, with an average of 1830.2 deaths per year. The spatial analysis in the 399 municipalities of Parana revealed that the LC mortality rates ranged between 21.0 and 84.2 deaths per 100,000 inhabitants from 2015 to 2019 (Fig 2A).

Global Spatial Autocorrelation Index analysis allowed the identification of significant positive spatial autocorrelation in the entire region studied (I = 0.892, p = 0.001), demonstrating that the LC mortality is not randomly distributed. The LISA analysis indicated that the patterns of LC mortality in Parana state were irregular and there were two big clusters. A total of 106 municipalities in the southern region of the state presented direct (positive) spatial correlation with high LC mortality (hot spots or high-high clusters) and 119 municipalities in the north showed low LC mortality (cold spots or low-low clusters) (Fig 2B).

The OLS regression results showed that two variables (tobacco production rate and accessibility to oncology centers by municipality) were directly related to LC mortality (t-value higher than 1.96) and two variables (disease diagnosis rate and IPDM) were inversely related to LC mortality (t-value lower than -1.96. Moreover, a significant improvement (lower Moran's I

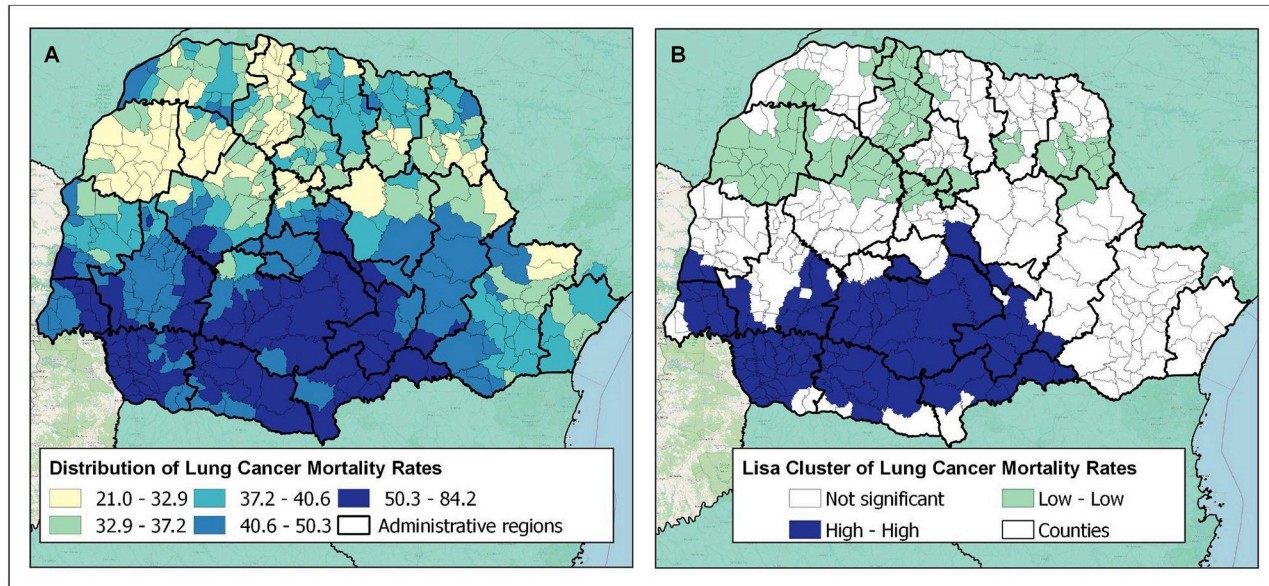

**Fig 2. Spatial distribution of the lung cancer mortality (LC mortality) rate, between 2015 and 2019, in Parana, Brazil.** A) LC mortality rates per 100,000 inhabitants (age-adjusted); B) location of high-high and low-low clusters according to LC mortality rates in Parana.

residue and AIC, and higher $R^2$) could be seen in the GWR compared to OLS, indicating a better overall adjustment of the local spatial model in comparison with the global model (Table 1).

The results of the GWR analysis showed the distinct spatial impact and patterns of each variable associated with LC mortality. Tobacco production showed only a positive significant correlation with LC mortality (in green) (Fig 3A), while the other variables (Fig 3B–3D) showed positive (in green) association in some places and negative in others (in blue).

**Table 1. Comparison of the Ordinary Least Squares (OLS) and Geographically Weighted Regression (GWR) multivariate spatial regression models.**

| Variable | OLS | | | | GWR |
|---|---|---|---|---|---|
| | Coefficient | SD[a] | t-value | p-value | |
| Constant | 48.044 | 5.006 | 9.60 | <0.001 | |
| TPR[b] | 0.005 | $0.04 \times 10^{-4}$ | 11.27 | <0.001 | |
| DDR[c] | -828.215 | 319.097 | -2.60 | 0.010 | |
| AOC[d] | 0.606 | 0.123 | 4.78 | <0.001 | |
| IPDM[e] | -16.107 | 7.593 | -2.12 | 0.035 | |
| AIC[f] | - | 2902 | - | - | 2519 |
| Adjusted $R^2$ | - | 0.29 | - | - | 0.77 |
| Sum of residual squares | - | 32675 | - | - | 7887 |
| Moran's I (residual) | - | 0.57 | - | - | 0.27 |

[a]standard-deviation,

[b]tobacco production rate,

[c]disease diagnosis rate,

[d]accessibility to oncology centers by municipality,

[e]Parana Municipal Performance Index, and

[f]Akaike Information Criterion.

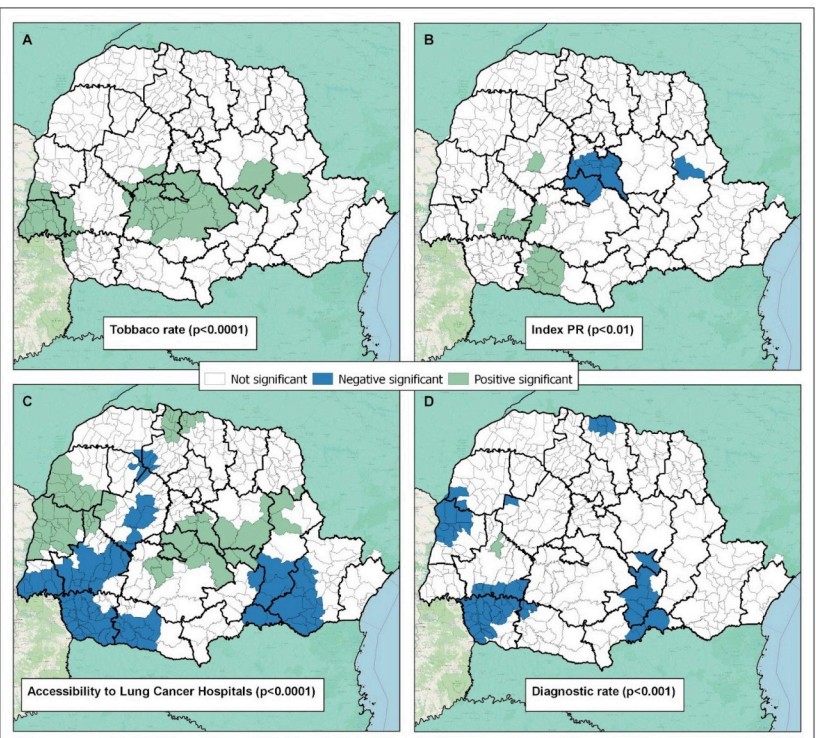

**Fig 3. Tobacco production rate (A), Parana Municipal Performance Index (B), Accessibility to Oncology Centers by Municipality (C), Disease diagnosis rate (D), spatial patterns in the state of Parana, Brazil, according to Geographically Weighted Regression (GWR) Analysis.**

## Discussion

Geography and health are intrinsically linked. The location of our housing and workplaces influences our health experiences, from the air we breathe and the food we eat, to the viruses that we are exposed to and the health services we can access [39]. The state of Parana is a state that, for the most part, is made up of municipalities with rural characteristics with less than 40,000 inhabitants, representing about 88.7% of the total number of municipalities (354/399) [40]. It is also a major producer and exporter of tobacco [41]. Therefore, the investigation of LC Mortality and tobacco production using geospatial analyzes is very important not only for understanding the epidemiological situation but also for helping to take adequate protective measures and treat the affected population.

Since the implementation of anti-smoking policies in the late 1980s, the prevalence of LC has been reduced by almost half in the Brazilian population. However, LC is still one of the main causes of cancer deaths in Brazil [8]. This information highlights the importance of analyzing the correlation between LC mortality and spatial distribution of its indicators. Informed decision-making leads to the development of effective public health policies that are based on evidence and geographical analysis can help to identify potential exposure that deserve epidemiological investigation [39]. As far as we know this is the first study to investigate the association between LC mortality and tobacco production rate, IPDM, accessibility of oncology centers by municipality, and disease diagnosis rate.

We observed a negative association between LC mortality and the Parana Municipal Performance Index, as well as disease diagnosis rate. On the contrary, the tobacco production rate

(TPR) and accessibility to oncology centers showed a positive association with LC mortality. In the 2020/2021 harvest, Parana had a share of 24.7% (R$ 1,637,382,686.00) in the gross revenue of tobacco in the southern region, thus raising, at the national level, the percentage of production from 5% to 25% in 2021. Currently, 65,200 hectares of soil in Parana are used for tobacco cultivation, with a production of 147,400 thousand tons, with a productivity of 2,260 kg per hectare [42].

The association between tobacco production and LC mortality in the central, southern, southwest and western mesoregions of Parana state [38, 39] is probably related to the tobacco production process [1], since these regions are the biggest tobacco producers [43, 44]. In Brazil, *Nicotiana tabacum* is usually harvested and its leaves manipulated by producers, often without any personal precautions, exposing them to pesticides and polycyclic aromatic hydrocarbons, increasing the probability of LC development and mortality [45–48].

Other factors, such as living in rural areas, low socioeconomic conditions or limited education can lead to inadequate screening of new cases of LC and delayed access to medical services, thereby increasing the LC mortality [49, 50]. Reciprocally, this means that scientific and technological advances, specifically early screening and improved diagnostics and treatment, can reduce LC mortality.

A previous study in the United States (US) showed that a person who lives in the rural areas, where levels of education are low and there is difficulty in accessing medical services (i.e.: chemotherapy, radiotherapy and surgery) has a higher risk of death due to LC compared to urban residents [46]. Another study in the US describes that the people living in the rural area and/or with limited education, had relatively less access to chest tomography facilities, making it difficult to track the LC in this population [51].

Despite considerable development in LC surgery in the last 10 years, specifically minimally invasive techniques, access to standard or advanced health services within the Brazilian Health System is also challenging. Access to specialized free services is very difficult, and some tests and therapies, especially the new ones, are scarce or not available in the public network, thereby causing extensive delays in the care of patients with this type of cancer [52].

In our study, we found that municipalities with higher accessibility to oncology centers had a higher LC mortality. This could be due to the fact that increases in specialized health services utilization can optimize LC diagnosis thereby demonstrating more appropriate reporting [8]. In the large and highly populated cities, due to high patient demand for health services, it can be difficult to schedule appointments with specialists.

Another important parameter used to measure the socioeconomic conditions in the state of Parana is the IPDM (Parana Municipal Performance Index). This parameter is composed of the three dimensions of human development: 1) income, employment and agricultural production, 2) health, and 3) education [52]. Our study also revealed a negative association between LC mortality and IPDM, confirming that less developed areas tend to have high LC mortalities [8, 15, 49]. Our results mirror previous studies that have revealed that living in rural areas with low levels of education and income is associated with a lack of chemotherapy prescription, radiotherapy and surgery, resulting in higher mortality rates.

Our results showed that lower IPDM is related with higher risk of death due to LC. According to the literature, there is an association between IPDM and LC occurrence, since higher educational level, occupation, and income, usually reflect in lower risk of LC occurrence and deaths [53].

Overall, LC occurrence and deaths are multifactorial, and a myriad of factors including ethnicity and gender, can change this rate. As such when evaluating only one aspect of LC mortality, we could easily misinterpret the results. Individuals with low education and income levels

have a higher risk of LC [50], since they are usually the main producers of tobacco, and, in most cases, do not have enough knowledge about the problems associated with it [45–48].

## Study limitations

The use of secondary data has some limitations, such as the possibility of underreporting. Another possible limitation is the fact that our data is only analyzed and interpreted considering the Parana state. Social, economic and cultural conditions can change across Brazilian regions. Considering that these differences can influence the IPDM and specialist/oncologist treatment and availability/treatment seeking amongst other factors, these results might not be generalizable to other regions of Brazil.

## Conclusion

The partnership among health specialists and geospatial analysis can help sustain innovative approaches to solving complex problems and ultimately reduce inequity in healthcare. In this research there was a significant positive spatial autocorrelation in the Parana municipalities, indicating that the LC mortality is not randomly distributed, with the south of the state having a direct (positive) spatial correlation with a high LC mortality rate. Risk factors for higher mortality by LC rates are not restricted to socioeconomic conditions, but also to risky behaviors such as *Nicotiana tabacum* harvesting or the accessibility to specialized health services.

The issues facing health and healthcare are complex and an integrative and multidisciplinary approach is crucial to ensure that research provides relevant, high-quality evidence to inform health policy. Since social, economic and cultural conditions can change across Brazilian regions, influencing the LC mortality, it is important to develop complementary studies to evaluate the factors associated with it in other regions, using datasets from the other states to better understand the profile of LC mortality in Brazil.

## Acknowledgments

We thank the Health Technologies and Geoprocessing Group (Health Technologies and Geoprocessing Group—GETS) for their support in the geospatial analysis.

## Author Contributions

**Conceptualization:** Vlaudimir Dias Marques, Mariana Teixeira da Silva, Igor Roskowski, Daniel Augusto Nunes de Lima, Lander dos Santos, Estela Louro, Simone Tomás Gonçalves, Raissa Bocchi Pedroso, Amanda de Carvalho Dutra, Luciano de Andrade.

**Formal analysis:** Amanda de Carvalho Dutra, Luciano de Andrade.

**Investigation:** Luciano de Andrade.

**Methodology:** Vlaudimir Dias Marques, Amanda de Carvalho Dutra, Luciano de Andrade.

**Project administration:** Vlaudimir Dias Marques, Luciano de Andrade.

**Supervision:** Luciano de Andrade.

**Validation:** Amanda de Carvalho Dutra, Luciano de Andrade.

**Visualization:** Amanda de Carvalho Dutra, Luciano de Andrade.

**Writing – original draft:** Vlaudimir Dias Marques, Miyoko Massago, Mariana Teixeira da Silva.

**Writing – review & editing:** Miyoko Massago, Raissa Bocchi Pedroso, Armstrong Mbi Obale, Sandra Marisa Pelloso, João Ricardo Nickenig Vissoci, Catherine Ann Staton, Oscar Kenji Nihei, Maria Dalva de Barros Carvalho, Luciano de Andrade.

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
