## [Decision Letter · Decision Letter 0]

3 Mar 2023

PONE-D-22-22860EXPLORING REGIONAL DISPARITIES IN LUNG CANCER MORTALITY IN A BRAZILIAN STATE: A CROSS-SECTIONAL ECOLOGICAL STUDYPLOS ONE

Dear authors,

Thank you for submitting your manuscript to PLOS ONE. After careful consideration, we feel that it has merit but does not fully meet PLOS ONE’s publication criteria as it currently stands. Therefore, we invite you to submit a revised version of the manuscript that addresses the points raised during the review process.

We look forward to receiving your revised manuscript.

Kind regards,

Fernanda Penido Matozinhos, Ph.D

Academic Editor

PLOS ONE

Journal Requirements:

The present study was carried out with the support of the Coordination for the Improvement of Higher Education Personnel Brazil - CAPES - Financial Code 001. We thank the Health Technologies and Geoprocessing Group (Health Technologies and Geoprocessing Group - GETS) for their support in the geospatial analysis.

4. We note that Figures 1 to 3 in your submission contain [map/satellite] images which may be copyrighted. All PLOS content is published under the Creative Commons Attribution License (CC BY 4.0), which means that the manuscript, images, and Supporting Information files will be freely available online, and any third party is permitted to access, download, copy, distribute, and use these materials in any way, even commercially, with proper attribution. For these reasons, we cannot publish previously copyrighted maps or satellite images created using proprietary data, such as Google software (Google Maps, Street View, and Earth). For more information, see our copyright guidelines: http://journals.plos.org/plosone/s/licenses-and-copyright.

a. You may seek permission from the original copyright holder of Figures 1 to 3 to publish the content specifically under the CC BY 4.0 license.  

Additional Editor Comments:

Dear Editor and authors,

Thank you for the opportunity to review this manuscript. I am grateful for the invitation.

After careful consideration, I feel the manuscript explores a very important topic and we invite you to submit a revised version, on specific issues of the manuscript that addresses the points raised during the review process. The objective, the design and the results of this study are relevant.

Kind regards,

Reviewers' comments:

Reviewer's Responses to Questions

**Comments to the Author**

1. Is the manuscript technically sound, and do the data support the conclusions?

Reviewer #1: Yes

Reviewer #2: Yes

2. Has the statistical analysis been performed appropriately and rigorously? 

Reviewer #1: Yes

Reviewer #2: Yes

3. Have the authors made all data underlying the findings in their manuscript fully available?

Reviewer #1: No

Reviewer #2: Yes

4. Is the manuscript presented in an intelligible fashion and written in standard English?

Reviewer #1: Yes

Reviewer #2: Yes

5. Review Comments to the Author

Reviewer #1: The paper explores the relationship between lung cancer mortality and some variables, regarding a brazilian region. The procedures and techniques used seem sound and the conclusion follows from the results´ analysis. It is important to make a final revision of the text and to make the whole set of data available to the audience putting all of it in one place (in the web).

Reviewer #2: The study presents an interesting spacial description and analysis of lung cancer cases in Parana. This kind of study is important to better destine public policies in the state. However, I miss some things in the discussion: what are the study strengths? And what are the health implications?

What the managers can do with this information? For me, this study it's more important for this public, so they need to understand what is the scenario and what they can do.

Minor points

The title is duplicated in the submission.

Figures 2 and 3: The caption should be below the figure, not in the title. Figure 3 needs a title. The sentence there is the caption.

Line 302-304: Why these areas? Some people that will ready your work don't know Parana. I miss some explanation or hypothesis for this. It's because there are rural areas? Or because tobacco production occurs in these regions?

6. PLOS authors have the option to publish the peer review history of their article (what does this mean?). If published, this will include your full peer review and any attached files.

Reviewer #1: No

Reviewer #2: **Yes: **Luana Lara Rocha

---

## [Author Response · Author response to Decision Letter 0]

4 May 2023

Dear Fernanda Penido Matozinhos

Editor in chief of PloS One

We are sending the detailed response to the editor and reviewer’s comments to our manuscript entitled “Exploring regional disparities in lung cancer mortality in a Brazilian state: a cross-sectional ecological study”.

In the following table, we described how each editor and reviewer’s comment, sent in the revision of manuscript, was addressed.

Editor’s comments

Comment 1. 

We note that Figures 1 to 3 in your submission contain [map/satellite] images which may be copyrighted. All PLOS content is published under the Creative Commons Attribution License (CC BY 4.0), which means that the manuscript, images, and Supporting Information files will be freely available online, and any third party is permitted to access, download, copy, distribute, and use these materials in any way, even commercially, with proper attribution.

For these reasons, we cannot publish previously copyrighted maps or satellite images created using proprietary data,

such as Google software (Google Maps, Street View, and Earth). For more information, see our copyright guidelines: http://journals.plos.org/plosone/s/licenses-and-copyright.

Response: 

We would like to clarify that these figures were generated using the Brazilian shapefile that was obtained from the Brazilian Institute of Geography and Statistics (IBGE), freely available at: https://www.ibge.gov.br/geociencias/organizacao-do-territorio/malhas-territoriais/15774-malhas.html?=&t=downloads, in the archive “PR_municipios_2020.zip”.

In addition, in relation to Figure 1, part of it was generated using the South America shapefile, obtained freely at: https://tapiquen-sig.jimdo.com/english-version/free-downloads/south-america/, and the open street map, obtained freely using a plugin “open street map” of software QGIS.

After obtaining these shapefile archives, the final maps with the spatial analysis results were produced using the QGIS software as described in the article´s Methods Section. This information were further clarified in the article´s Methods Section and in the

legend of figures.

Reviewer 1

Comment 1

The paper explores the relationship between lung cancer mortality and some variables, regarding a Brazilian region. The procedures and techniques used seem sound and the conclusion follows from the results´ analysis. It is important to make a final revision of the text and to make the whole set of data available to the audience putting all of it in one place (in the web)

Response

All the text was reviewed as requested and the whole set of data was made available to audience in the figshare website: https://figshare.com/s/f6a8509b272a8c247a30

Reviewer 2

Comment 1

The study presents an interesting spatial description and analysis of lung cancer cases in Parana. This kind of study is important to better destine public policies in the state. However, I miss some things in the discussion: what are the study strengths? And what are the health implications? What the managers can do with this information? For me, this study it's more important for this public, so they need to understand what is the scenario and what they can do.

Response

We have reviewed and added this information more clearly in the discussion and conclusion sections as requested.

Comment 2

The title is duplicated in the submission.

Response

We excluded the duplicated title.

Comment 3

Figures 2 and 3: The caption should be below the figure, not in the title. Figure 3 needs a title. The sentence there is the caption.

Response

We are proceeding according to the journal's submission guidelines. Figure images are submitted separately from the main text of the manuscript. Figure legends are submitted within the text of the manuscript. The final format is presented according to the submission system settings.

Comment 4

Line 302-304: Why these areas? Some people that will ready your work don't know Parana. I miss some explanation or hypothesis for this. It's because there are rural areas? Or because tobacco production occurs in these regions?

Response 

The discussion and conclusion sections were reviewed and more information was added to clarify our findings as requested.

We look forward to further editorial orientations,

Best Regards

Vlaudimir Dias Marques

Corresponding Author

---

## [Decision Letter · Decision Letter 1]

6 Jun 2023

EXPLORING REGIONAL DISPARITIES IN LUNG CANCER MORTALITY IN A BRAZILIAN STATE: A CROSS-SECTIONAL ECOLOGICAL STUDY

PONE-D-22-22860R1

Dear,

We’re pleased to inform you that your manuscript has been judged scientifically suitable for publication and will be formally accepted for publication once it meets all outstanding technical requirements.

Kind regards,

Fernanda Penido Matozinhos, Ph.D

Academic Editor

PLOS ONE

Additional Editor Comments (optional):

Dear authors, the manuscript explores a very important topic and it has technical rigor.

Thank you for submitting your manuscript to PLOS ONE and making substantial changes in order to improve the manuscript. I congratulate the authors for the work developed. The objective is relevant and the results are of interest for a wide range of potential readers. I recommend its publication.

Reviewers' comments:

Reviewer's Responses to Questions

**Comments to the Author**

1. If the authors have adequately addressed your comments raised in a previous round of review and you feel that this manuscript is now acceptable for publication, you may indicate that here to bypass the “Comments to the Author” section, enter your conflict of interest statement in the “Confidential to Editor” section, and submit your "Accept" recommendation.

Reviewer #1: All comments have been addressed

Reviewer #2: All comments have been addressed

2. Is the manuscript technically sound, and do the data support the conclusions?

Reviewer #1: (No Response)

Reviewer #2: Yes

3. Has the statistical analysis been performed appropriately and rigorously? 

Reviewer #1: (No Response)

Reviewer #2: Yes

4. Have the authors made all data underlying the findings in their manuscript fully available?

Reviewer #1: (No Response)

Reviewer #2: Yes

5. Is the manuscript presented in an intelligible fashion and written in standard English?

Reviewer #1: (No Response)

Reviewer #2: Yes

6. Review Comments to the Author

Reviewer #1: (No Response)

Reviewer #2: (No Response)

7. PLOS authors have the option to publish the peer review history of their article (what does this mean?). If published, this will include your full peer review and any attached files.

Reviewer #1: No

Reviewer #2: No

---

## [Editor Report · Acceptance letter]

13 Jun 2023

PONE-D-22-22860R1 

EXPLORING REGIONAL DISPARITIES IN LUNG CANCER MORTALITY IN A BRAZILIAN STATE: A CROSS-SECTIONAL ECOLOGICAL STUDY 

Dear Dr. Dias Marques:

I'm pleased to inform you that your manuscript has been deemed suitable for publication in PLOS ONE. Congratulations! Your manuscript is now with our production department. 

Kind regards, 

on behalf of

Dr. Fernanda Penido Matozinhos 

Academic Editor

PLOS ONE